# Aesthetic dental treatment, orofacial appearance, and life satisfaction of Finnish and Brazilian adults

Lucas Arrais Campos[1,2,3,4]*, Juliana Alvares Duarte Bonini Campos[5], João Marôco[6,7], Timo Peltomäki[1,2,3,8]

**1** Faculty of Medicine and Health Technology, Tampere University, Tampere, Finland, **2** Department of Ear and Oral Diseases, Tampere University Hospital, Tampere, Finland, **3** Faculty of Health Sciences, Institute of Dentistry, University of Eastern Finland, Kuopio, Finland, **4** School of Dentistry, Campus Araraquara, São Paulo State University (UNESP), São Paulo, Brazil, **5** School of Pharmaceutical Sciences, São Paulo State University (UNESP), São Paulo, Brazil, **6** William James Center for Research (WJCR), ISPA-Instituto Universitário, Lisbon, Portugal, **7** Flu Pedagogy, Nord University, Bodø, Norway, **8** Department of Oral and Maxillofacial Diseases, Kuopio University Hospital, Kuopio, Finland

* lucas.arraisdecampos@tuni.fi

**Data Availability Statement:** All relevant data are within the paper and its Supporting information files.

## Abstract

### Aims

To study the probability of seeking/undergoing aesthetic dental treatment (ADT) and compare self-perception of orofacial appearance (OA) based on sex, age, and monthly income; and to estimate the impact of OA on life satisfaction (LS) among Finnish and Brazilian adults, considering the indirect effect of receiving ADT and the moderating effects of those sociodemographic variables.

### Methods

This was an online cross-sectional study. Orofacial Esthetic Scale (OES), Psychosocial Impact of Dental Aesthetics Questionnaire (PIDAQ) and Satisfaction with Life Scale (SWLS) were used. Probability of seeking/receiving ADT was calculated using logistic regression and odds ratio (OR). OA scores were compared according to sociodemographic characteristics (ANOVA, α = 5%). Structural equations models estimated the impact of OA on LS.

### Results

3,614 Finns [75.1% female, 32.0 (SD = 11.6) years] and 3,979 Brazilians [69.9% female, 33.0 (SD = 11.3) years] participated in the study. Women were more likely to receive ADT than men in both countries (OR>1.3). However, no statistically or practical significant differences were observed in OA between sexes (p>0.05 or p<0.05, $\eta_p^2$ = 0.00–0.02). In Finland, demand for ADT (OR = 0.9–1.0) and OA scores (p>0.05) were the same among different ages and monthly income. In Brazil, younger individuals (OR>1.6) and those with higher monthly income (OR>2.7) were more likely to receive ADT, while those with lower income had a greater psychosocial impact of OA (p<0.05; $\eta_p^2$>0.07). Individuals who were more

**Funding:** This study received financial support from São Paulo Research Foundation (FAPESP) (grant number#2018/06739-1) awarded to LAC and (grant number #2019/19590-9) awarded to JADBC. This study was financed in part by the Coordenação de Aperfeiçoamento de Pessoal de Nível Superior - Brasil (CAPES) (grant number 001), awarded to LAC. This study was partly financially supported by the State funding for university-level health research, Tampere University Hospital, Wellbeing services county of Pirkanmaa, Finland (grant number 9AC074) awarded to TP. The funders had no role in study design, data collection and analysis, decision to publish, or preparation of the manuscript.

**Competing interests:** The authors have declared that no competing interests exist.

satisfied with their own OA and had less psychosocial impact from OA had higher levels of LS (β = 0.31–0.34; p<0.01; explained variance: 9.8–13.1%).

## Conclusion

Demand for ADT is influenced by sociodemographic and cultural factors. Greater societal pressure on physical appearance is observed among women in Western countries. In countries with high socioeconomic inequalities, consumerism and social prestige are involved in this demand. Self-perception of orofacial appearance plays a significant role in individuals' subjective well-being. Therefore, the planning of aesthetic treatments in the orofacial region should consider the patient's perceptions and social context.

## Introduction

Since ancient times, societies have valued physical appearance, which today remains to be an important characteristic that can affect various aspects of an individual's life [1, 2]. Orofacial appearance (teeth and face) is a notable feature of physical appearance and is strongly related to interpersonal relations [1]. This is because orofacial region plays a large role in the process of communication, identification, and self-identity construction [1, 3, 4]. From the orofacial appearance, impressions are also quickly formed regarding an individual's personality and moral and social characteristics [4, 5]. Despite their limited accuracy, these impressions have a place in social behavior and routine decision making, resulting in privileges or disadvantages based on orofacial appearance [5]. An individual who is aware of this can then adopt body-altering behaviors, aiming to obtain a good-looking appearance based on self-perception and socially established standards [1]. Undergoing aesthetic procedures [6], including aesthetic dental treatments, are among these behaviors.

With progress advancing in the field of aesthetic dental treatments, studies over the last two decades have stated a growing demand for these treatments [7–10]. Although this statement is widely recognized and reported in clinical dental practice, there are limited studies [9–12] that provide specific quantification of desire or demand for aesthetic dental treatments across different populations. Samorodnitzky-Naveh et al. [9] conducted a survey with 407 18-26-year-old dental patients in Israel, of whom 77.4% desired to improve their dental appearance. Wulfman et al. [10] found that 38.0% of French seniors sample (n = 3,868, age≥55 years) expressed a desire to change their smile, with women and younger part of the sample expressing a greater desire. In a study conducted on 31-year-old Brazilians (n = 536) in 2018, Silva et al. [11] found that 85.9% reported being interested in tooth whitening treatment.

This finding by Silva et al. [11] is similar to that by Campos et al. [12] in 2022, in which study 81.9% of a Brazilian general population sample (age≥18 years, n = 1,468) reported having sought aesthetic dental treatment. In the same study [12], a Finnish sample (n = 3,636, age ≥18 years) was also investigated: less than half (40.6%) reported having sought such a treatment. It seems that demand for aesthetic dental treatment is influenced by cultural [12] and sociodemographic factors [9, 10] such as sex, age, and economic level, and should be taken into account when the demand is scrutinized.

It is also important to consider that demand refers to a behavior adopted based on an individual's perspectives and perceptions. In dentistry, self-perception of orofacial appearance stands out [1, 12], being one of the main dimensions of oral health-related quality of life (OHRQoL) and can represent a reason why dental treatment is sought [13, 14]. Therefore,

including the self-perception of orofacial appearance in research on demand for aesthetic dental treatment is important. Because the dimensions of OHRQoL cannot be directly measured, the use of specific means, psychometric scales, are necessary [14, 15]. The Orofacial Esthetic Scale (OES) [15–17] and the Psychosocial Impact of Dental Aesthetic Questionnaire (PIDAQ) [15, 18, 19] are two scales that evaluate self-perception of orofacial appearance and have shown good indicators of validity and reliability in different populations.

Estimating the impact of the orofacial appearance dimension of OHRQoL on well-being of individuals with different cultural backgrounds is another important point to be considered. This information may be useful not only for advancing scientific evidence and contributing to the formation of professionals with a more holistic view of their patients [1], but also for fostering discussion about the social role of dentistry. Campos et al. [1] observed in a sample of Brazilian individuals aged 18 to 40 (n = 1,940) that orofacial appearance, measured by OES and PIDAQ, explained 9.9 to 14.3% of the variance in life satisfaction (cognitive aspect of subjective well-being). Although the orofacial appearance occupies a prominent space in an individual's life [1, 12–14], to the best of our knowledge no other studies have evaluated their direct contribution to subjective well-being in general populations, which would be relevant for the development of a patient-centered treatment plan [1].

Self-perception of orofacial appearance may vary according to different sociodemographic characteristics, such as sex, age, and socioeconomical level [16, 20, 21]. These characteristics may therefore also have an effect on how orofacial appearance impacts subjective well-being. Thus, it is relevant to investigate the differences in orofacial appearance according to sociodemographic characteristics and evaluate their moderating role on the relationship between orofacial appearance and well-being. Cultural and social values can also influence the role of these sociodemographic variables in the demand for aesthetic dental treatment and the perception of orofacial appearance, as well as its impact on well-being. Therefore, to identify coherences and specificities, it is worthwhile to extend this investigation to countries with significant sociocultural differences initially.

Finland and Brazil are examples of countries with such sociocultural discrepancies. Finland has one of the closest levels of gender equality [22], low inequality between different socioeconomic classes [23], and similar living conditions of its population. Brazil, on the other hand, has high inequality and different living conditions among different sociodemographic groups [22, 23], which affects access to health treatments, especially aesthetic dental treatment, since it is provided in the private sector. Moreover, the value attributed to physical appearance varies between these countries, with physical appearance carrying much more importance in social interactions and behaviors for Brazilians [12]. Thus, studying both countries simultaneously is a good starting point for cross-national comparisons.

The objectives of this study were 1. to study the probability of Finnish and Brazilian adults of seeking and undergoing aesthetic dental treatment according to sex, monthly income, and age, 2. to compare the self-perception of orofacial appearance in Finland and Brazil according to sex, monthly income, and age, and 3. to estimate the impact of self-perception of orofacial appearance on life satisfaction in Finnish and Brazilian adults, taking into consideration the indirect effect of receiving aesthetic dental treatment and the moderating effects of sex, monthly income, and age on this impact.

## Methods

### Study design and sampling

This was a cross-sectional study with snowball non-probability sample selection. Finnish and Brazilian individuals over the age of 18 years were invited to participate in the study. The

selection of Brazil and Finland for the study was based on their sociocultural differences, as well as the convenience of the researchers whose work is located in these countries. Initially the invitation was sent to students and staff from universities in Finland and Brazil. Then, snowball strategy was used to recruit more participants. Because the data were collected during the pandemic, this sampling strategy was the most feasible to address the aims of the study.

The minimum sample size was calculated following the proposal by Hair et al. [24], who recommend a minimum of 10 participants per parameter to be estimated in the structural model. In the present study, 28 parameters were considered a priori to be estimated in the model. Thus, the minimum sample size required for each country was 280 participants. However, a larger number of participants were recruited to increase the variability and coverage of the data for the study populations.

Demographic information was collected on sex (male, female, or other/not informed), age, marital status (single, married/common law/stable relationship, divorced, widower), monthly income, and whether the individual has sought or received any aesthetic dental treatment (no, yes). Although age was collected in years it was categorized according to the 25th, 50th, 75th, and 90th percentiles when considering the samples from both countries simultaneously (1: <23 years, 2: 23 ⊢ 29 years, 3: 29 ⊢ 39 years, 4: 39 ⊢ 52 years, and 5: ≥52 years). The monthly income was collected based on information from Statistics Finland [25] and *Centro de Políticas Sociais*–FGV Social (Brazil) [26] and was stratified into the following categories: Finland– 1: <2,500 €, 2: 2,500 ⊢ 5,000 €, 3: 5,000 ⊢ 7,500 €, 4: 7,500 ⊢ 10,000 €, 5: ≥10,000 € Brazil– 1: <R$ 1,255, 2: R$ 1,255 ⊢ 2,005, 3: R$ 2,005 ⊢ 8,641, 4: R$ 8,641 ⊢ 11,262, 5: ≥R$ 11,262.

## Procedures and ethical aspect

Invitation message was sent to individuals by institutional email. The message contained information regarding the aims of the study, ethical approval, and a link to the online survey which was created using the LimeSurvey software (LimeSurvey GmbH, Hamburg, Germany; URL http://www.limesurvey.org) on the server of Tampere University, Finland. To start the online survey, the participants were informed that the responses were anonymous and had to agree and give written informed consent. At first, the demographic questionnaire was presented followed by the measuring scales in random order. At the end of the survey, the participants were asked to forward the invitation message and survey link to their contacts via email or social media (snowball sampling). The data was collected between June and July 2020 in Finland and June 2020 and March 2021 in Brazil. In Brazil, the link had to remain open longer because the participants' adhesion to the survey was slower than in Finland. Initially, larger sample was planned in Brazil because of larger population than Finland. However, even after 9 months, we were only able to have a similar sample size in the two countries.

This study was approved by the Data Protection Officer at Tampere University, in accordance with the European Union's General Data Protection Regulation, and by the Research Ethics Committee of São Paulo State University (Unesp), School of Dentistry, Araraquara (CAAE: 88600318.3.0000.5416). In both countries, participation in the study was voluntary and anonymous, and participants did not receive any incentives to take part. The authors had no access to information that could identify individual participants during or after data collection.

## Measuring scales

The self-perception of orofacial appearance was assessed using the Finnish and Portuguese versions of the Orofacial Esthetic Scale (OES) [15–17] and Psychosocial Impact of Dental Aesthetic Questionnaire (PIDAQ) [15, 18, 19]. The OES is a 7-item, one-dimension scale with an

11-point numerical response scale ranging from 0 (very dissatisfied) to 10 (very satisfied). This scale assesses the satisfaction with specific orofacial physic aspects. The OES also has an eighth item that assesses the satisfaction with the overall appearance. However, this item is not considered in the factor model nor for calculating the mean score as suggested by the authors proposing this scale [17].

Version of PIDAQ presented by Campos et al. [15, 20] was used in the present study. This version has 24 items with 5-point Likert-type response scale (0: I do not agree to 4: I totally agree) that assess 4 dimensions of the psychosocial impact of dental aesthetics (Dental Self-Confidence, Social Impact, Psychological Impact, and Aesthetic Concern). The need for exclusion of 4 items in Social Impact dimension for Finnish sample and 1 item in Social Impact dimension for Brazilian sample was observed in previous studies [12, 15] estimating the psychometric properties of PIDAQ. Therefore, these items were not considered in the factor model and for calculating the mean score for each sample in the present study.

The subjective well-being was assessed using the Satisfaction with Life Scale (SWLS) [27–29]. It consists of 5 items with 7-point Likert-type response scale (1: strongly disagree to 7: strongly agree) and measures one dimension related to the individual's overall life satisfaction.

## Data validity and reliability

The validity of the data was verified using confirmatory factor analysis (CFA). To assess the psychometric sensitivity of the scales' items [30], the distribution of responses was estimated using measures of skewness (sk) and kurtosis (ku). Criteria for non-severe violations of univariate normality were defined as absolute values of sk < 3.0 and ku < 10 [31]. Multivariate normality for each scale's responses was assessed by calculating the ratio of multivariate kurtosis to critical ratios ($ku_m/cr$) [30]. Absolute values of $ku_m/cr$ less than 3 indicated multivariate normality [30].

The maximum likelihood (for OES) or the robust weighted least squares mean and variance adjusted (for PIDAQ and SWLS) estimation methods were used. The fit of the factor models to the data was considered adequate when the comparative fit index (CFI) and the Tucker-Lewis index (TLI) were both greater than 0.90, the root mean square error of approximation (RMSEA) < 0.10, the standardized root mean square residual (SRMR) < 0.08, and standardized factor loadings ($\lambda$) > 0.50 [30, 31]. The reliability of the data was assessed using Cronbach's alpha coefficient (for OES) or ordinal alpha coefficient (for PIDAQ and SWLS) and values > 0.70 were considered adequate [30, 31]. Measurement invariance was tested to verify whether it would be possible to compare the mean scores obtained using the scales between subsamples according to variables of interest (sex, monthly income categories, and age categories) [32]. Multigroup analysis considering CFI difference ($\Delta$CFI) between configural and metric models (metric invariance) and between metric and scalar models (scalar invariance) was performed between the subsamples of each country. Reductions in CFI smaller than 0.01 were indicative of measurement invariance. The analyses were conducted in the R program (R Core Team, 2022) using the "*lavaan*" [33] and "*semTools*" [34] packages.

The responses to the items within each scale demonstrated a distribution that closely approximated the normal distribution, as well as evidence of multivariate normality (Finland–OES: $|sk| \leq 1.2$, $|ku| \leq 1.9$, $ku_m/cr = 0.4$; PIDAQ: $|sk| \leq 2.6$, $|ku| \leq 6.5$, $ku_m/cr = 1.0$; SWLS: $|sk| \leq 1.1$, $|ku| \leq 1.0$, $ku_m/cr = 0.3$; Brazil–OES: $|sk| \leq 1.3$, $|ku| \leq 1.7$, $ku_m/cr = 0.4$; PIDAQ: $|sk| \leq 2.9$, $|ku| \leq 3.8$, $ku_m/cr = 1.1$; SWLS: $|sk| \leq 1.3$, $|ku| \leq 1.8$, $ku_m/cr = 0.3$). The fit of the factor models of the scales was adequate to the Finnish and Brazilian samples, attesting the validity and reliability of the data (S1 Table). The models also showed adequate fit to the subsamples data (CFI $\geq$ 0.94, TLI $\geq$ 0.90, RMSEA $\leq$ 0.13, SRMR $\leq$ 0.06, $\alpha \geq$ 0.82). Metric or

scalar measurement invariance was observed among the subsamples of each country according to sex, monthly income categories, and age category (S1 Table) making it possible to directly compare the mean scores [32].

## Data analysis

Descriptive analysis was performed to characterize the sample according to the country. The prevalence and 95% Confidence Interval (95%CI) of the participants who have sought and received aesthetic dental treatment were estimated and compared using z test ($\alpha$ = 5%) according to the sex, monthly income category, and age category. Logistic regression model was conducted separately according to the country and the odds ratio with 95%CI was calculated to verify the relationship of these sociodemographic variables with seeking and receiving aesthetic dental treatment. Sex (reference category (rc): male), monthly income category (rc: <2,500€/<R$1,255), and age category (rc: $\geq$52 years) were the independent variables. The reference category for the independent variables were established based on previous studies that identified the groups with the lowest prevalence of seeking/undergoing esthetic treatments [1, 9, 10, 12]. The dependent variable was having sought and received aesthetic dental treatment, separately.

The mean scores for the OES and PIDAQ factors were calculated for each participant considering the items that form the factorial model fitted to the sample data (S2 Table). The mean scores were compared according to sex, monthly income, and age category. The distribution of the scores in each group were estimated by skewness (Sk) and kurtosis (Ku). Absolute values of Sk and Ku lower than 3 and 10, respectively, were indicative of non-severe violation of normal distribution [31]. The scores showed no severe violation of the normal distribution (Sk$\leq$| 2.8|; Ku$\leq$|8.9|). The data homoscedasticity was evaluated using Levene's test. If data showed homoscedasticity, the scores were compared using ANOVA followed by Tukey post-hoc test. If heteroscedasticity was observed, the scores were compared using Welch's ANOVA followed by Games-Howell post-hoc test. The effect size of the difference between the groups was calculated using partial eta squared ($\eta_p^2$), and a significance level of 5% was adopted. The analysis was performed using IBM SPSS Statistics 28 (IBM Corp., Armonk, NY, USA).

To address the third aim of the study, i.e. to estimate the impact of self-perception of orofacial appearance on life satisfaction, structural equation analysis was conducted. Initially, the possibility of forming a single Orofacial Appearance dimension composed of the mean scores of the OES and PIDAQ factors was tested using Principal Component Analysis (PCA) with Promin rotation. The assumption of sampling adequacy for factoring was estimated by measures of sampling adequacy (MSA), with values higher than 0.7 considered adequate [24]. Data from Finland and Brazil meet the assumptions for the PCA (S2 Table). The number of factors to be retained in the PCA was determined by Parallel Analysis with random permutations of the observed data [35]. PCA and Parallel Analysis retained one factor (S3 Table) and for this reason, additionally, the suggestion of the unidimensionality of the scores was evaluated to confirm the adequacy of this proposed model. For this, the following indices and reference values were considered: Unidimensional Congruence (UniCo) > 0.95, Explained Common Variance (ECV) > 0.85, and Mean of Item Residual Absolute Loadings (MIREAL) < 0.30 [36]. Values of UniCo > 0.98, ECV > 0.89, and MIREAL < 0.27 were observed in Finnish and Brazilian sample. Therefore, these results suggest and confirm the possibility of treating the OES and PIDAQ factors scores as one dimension called Orofacial Appearance. PCA was performed using program Factor 11.05 for Windows [37].

In the structural model, the Orofacial Appearance dimension was considered as independent variable and the dimension of life satisfaction assessed by SWLS was the dependent

variable. The variable 'received aesthetic dental treatment' was inserted in the model as intermediate variable (indirect effect) between orofacial appearance and life satisfaction. The criteria for indirect effect were verified [38, 39] and bootstrap simulation analysis for Sobel's test was used for the evaluation of indirect effect path estimates [31]. Moderation analysis was conducted to estimate the moderating role of sex (1 = male, 2 = female), monthly income (Finland: 1 = <2,500 €, 2 = 2,500 ⊢ 5,000 €, 3 = 5,000 ⊢ 7,500 €, 4 = 7,500 ⊢ 10,000 €, 5 = ≥10,000 €; Brazil: 1 = <R\$ 1,255, 2 = R\$ 1,255 ⊢ 2,005, 3 = R\$ 2,005 ⊢ 8,641, 4 = R\$ 8,641 ⊢ 11,262, 5 = ≥R\$ 11,262), and age (years) between orofacial appearance and life satisfaction. First, CFA was conducted for the factor model of Orofacial Appearance, and factor scores were predicted from the factor score matrix obtained in the analysis [40]. Then, the interaction between factor scores and the moderation variables was added to the structural model. Also, following the theoretical rationale of the previous aims of the study, a direct path from sex, monthly income, and age to 'received aesthetic dental treatment' was added in the model.

The structural model elaborated is shown in Fig 1A and 1B. The fit of the model was considered adequate if CFI≥0.90, TLI≥0.90, RMSEA≤0.10 and SRMR≤0.08 [30, 31]. The significance of the hypothesized causal path estimates (β) was evaluated using the z-test (α = 5%), and the effect size was measured by the proportion of variance explained ($r^2$). The analysis was performed for each country separately in R program (R Core Team, 2020) using the "*lavaan*" [33] and "*semTools*" [34] packages.

## Results

A total of 3,614 Finns and 3,979 Brazilians participated in the study. The mean age of the Finnish participants was 32.0 (95%CI = 31.6–32.4) years and of the Brazilian participants was 33.0 (95%CI = 32.6–33.4) years. Table 1 shows the characteristics of the participants according to the country. In both samples, most of participants were women and single. In the Finnish sample, the majority (67.5%) had a monthly income in the two lower categories (up to 5,000 €), while in Brazil, a large majority (83.2%) had an income in the three higher categories (≥ R\$ 2,005). It may indicate differences in living conditions guarantees between the countries, with Brazil presenting greater inequalities in income distribution. In addition, it suggests that the income categories between the two countries are not directly comparable.

Most of the Finns have never sought or undergone a dental aesthetic treatment, while most of Brazilians have sought or undergone such treatment. Among those who sought aesthetic dental treatment, the prevalence of individuals who underwent such treatment was higher in the Finnish sample (94.8%, CI95% = 93.7–95.9%) than in the Brazilian sample (86.7%, CI95% = 85.5–87.9%), although both were high.

Table 2 and Fig 2 show the prevalence and probability, respectively, of seeking or receiving aesthetic dental treatment according to sex, monthly income, and age. For both countries, women were more prevalent and more likely to seek and receive this treatment than men. For Finland, no difference was found in seeking and receiving aesthetic dental treatment according to monthly income and age. For Brazil, younger people and those with higher monthly income had higher prevalence and chances of seeking and receiving such treatment. In general, differences between countries can be observed in Fig 2. While in Brazil, the OR is shifted to the left (OR>1.0) for different sociodemographic classes. In contrast, in Finland, the OR for age and economic level classes appears close to the alignment of the value 1.

The comparisons of dimensions scores of OES and PIDAQ according to sex, monthly income and age are presented in Table 3. For sex, although statistically significant differences were observed between men and women in the social and psychological impact dimensions (PIDAQ) in both countries, a low effect size was observed ($\eta_p^2$ = 0.002–0.022). This suggests

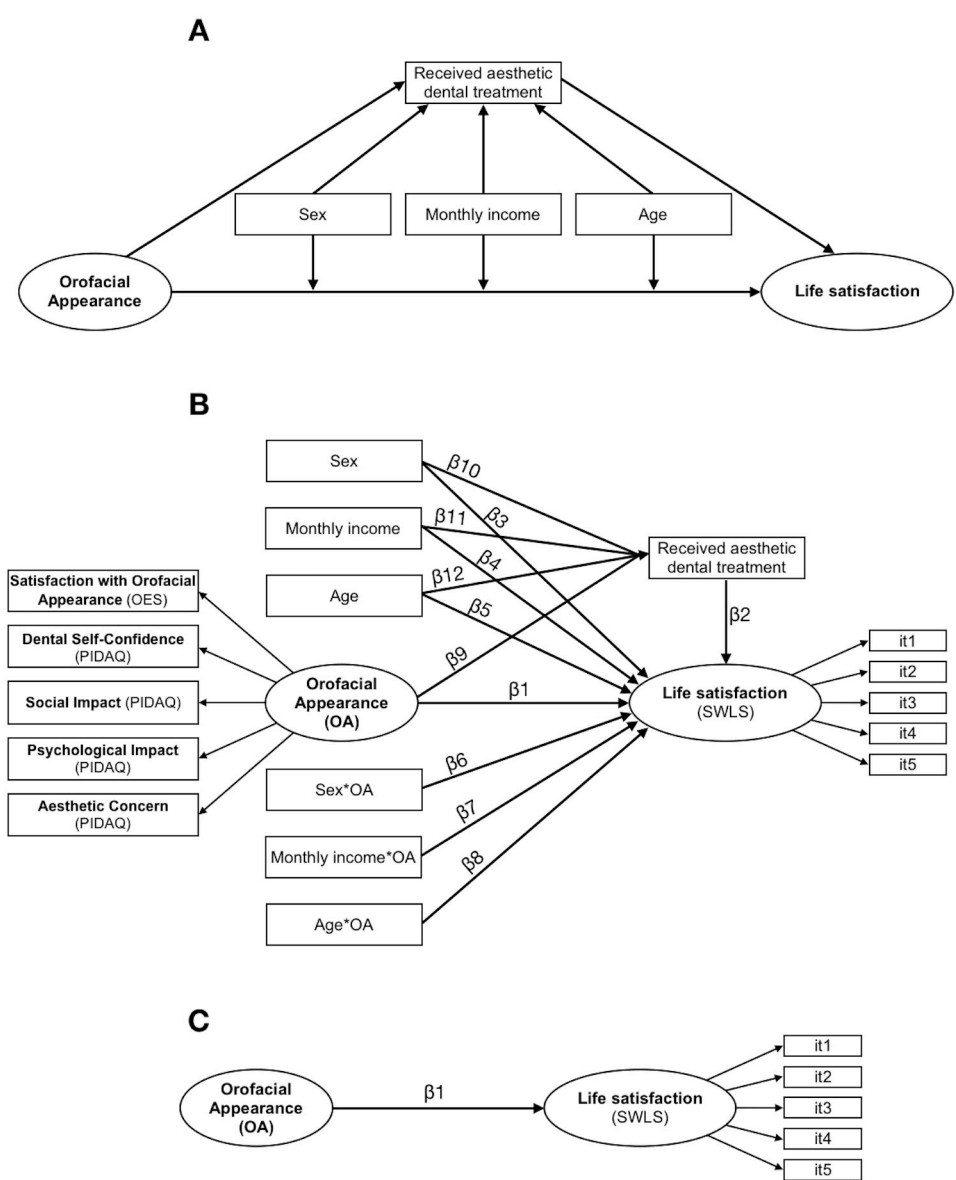

**Fig 1. Structural model elaborated to estimate the impact of orofacial appearance on life satisfaction the moderation role of sex, monthly income, and age, and the indirect effect of have received aesthetic dental treatment on this impact.** Note: **A:** Conceptual model. **B:** Statistical model. **C:** Refined model.

that the statistical significance may have been found by an inflation of the type I error due to a high sample size. It causes the statistical test to find very small differences with no practical significance between groups.

The same issue is observed in the age and monthly income variables. For both countries, statistical differences point to older individuals and those with higher monthly income being more satisfied with their orofacial appearance and less psychosocially affected by dental aesthetics. However, for most dimensions, these differences in scores between the age and monthly income categories were low and with small practical significance ($\eta_p^2$ = 0.007–0.042). An exception to this was the comparison of the scores of the social and psychological impact dimensions (PIDAQ) between the monthly income categories in the Brazilian sample. In these

**Table 1. Characteristics of the Finnish and Brazilian samples.**

| | | Sample (% (95%Confidence Interval)) | |
|---|---|---|---|
| **Characteristic** | | **Finnish (n = 3,614)** | **Brazilian (n = 3,979)** |
| **Sex** | | | |
| Male | | 23.3 (21.9–24.7) | 29.8 (28.4–31.2) |
| Female | | 75.1 (73.7–76.5) | 69.9 (68.5–71.3) |
| Other/Not informed | | 1.6 (1.2–2.0) | 0.3 (0.1–0.5) |
| **Marital status** | | | |
| Single | | 66.5 (65.0–68.0) | 59.8 (58.3–61.3) |
| Married/Common law/Stable relationship | | 28.8 (27.3–30.3) | 35.4 (33.9–36.9) |
| Divorced | | 4.5 (3.8–5.2) | 4.3 (3.7–4.9) |
| Widower | | 0.2 (0.1–0.3) | 0.5 (0.3–0.7) |
| **Monthly income** | | | |
| Finland (€) | Brazil (R$) | | |
| <2,500 | <1,255 | 44.6 (43.0–46.2) | 6.4 (5.6–7.2) |
| 2,500 ⊢ 5,000 | 1,255 ⊢ 2,005 | 22.9 (21.5–24.3) | 10.4 (9.5–11.3) |
| 5,000 ⊢ 7,500 | 2,005 ⊢ 8,641 | 14.3 (13.2–15.4) | 45.7 (44.2–47.2) |
| 7,500 ⊢ 10,000 | 8,641 ⊢ 11,262 | 8.7 (7.8–9.6) | 16.3 (15.2–17.4) |
| ≥10,000 | ≥11,262 | 9.5 (8.5–10.5) | 21.2 (19.9–22.5) |
| **Age category (years)** | | | |
| <23 | | 20.1 (18.8–21.4) | 21.1 (19.8–22.4) |
| 23 ⊢ 29 | | 34.2 (32.7–35.7) | 23.9 (22.6–25.2) |
| 29 ⊢ 39 | | 21.0 (19.7–22.3) | 28.1 (26.7–29.5) |
| 39 ⊢ 52 | | 15.0 (13.8–16.2) | 16.2 (15.1–17.3) |
| ≥52 | | 9.7 (8.7–10.7) | 10.7 (9.7–11.7) |
| **Have you ever sought any aesthetic dental treatment?** | | | |
| No | | 59.4 (57.8–61.0) | 22.1 (20.8–23.4) |
| Yes | | 40.6 (39.0–42.2) | 77.9 (76.6–79.2) |
| **Have you received any aesthetic dental treatment?** | | | |
| No | | 61.6 (60.0–63.2) | 32.5 (31.0–34.0) |
| Yes | | 38.2 (36.6–39.8) | 67.5 (66.0–69.0) |

cases, the difference in score between the first and last category was approximately 1.0 point, and the practical significance was considered medium ($\eta_p^2$ = 0.070–0.077).

The analyses of the structural models elaborated to estimate the impact of orofacial appearance on life satisfaction, the moderation role of sex, monthly income, and age, and the indirect effect of have received aesthetic dental treatment on this impact are shown in Table 4. The models did not present adequate fit to the sample (CFI≤0.64, TLI≤0.56, RMSEA≥0.195, SRMR≥0.278) and none of the demographic variables presented moderating effect (p≥0.10). It was also observed that there was no indirect effect of having received aesthetic dental treatment on the impact of orofacial appearance on life satisfaction (p>0.17). Therefore, the models were refined by excluding this variable, as well as those sociodemographic variables. The refined model presented adequate fit to the samples (Finnish Sample: CFI = 0.97, TLI = 0.95, RMSEA = 0.087, SRMR = 0.068; Brazilian Sample: CFI = 0.97, TLI = 0.96, RMSEA = 0.076, SRMR = 0.047). The orofacial appearance presented a significant impact on life satisfaction in both countries (Table 4). Individuals who are more satisfied with their own orofacial appearance and who perceive less of a psychosocial impact of dental aesthetic have higher life

**Table 2. Prevalence (% (95% Confidence Interval)) of the participants from Finland and Brazil who have sought and received aesthetic dental treatment in each category according to sex, monthly income, and age.**

| | | Finland | | | | Brazil | | | |
|---|---|---|---|---|---|---|---|---|---|
| | | Sought aesthetic dental treatment | | Received aesthetic dental treatment | | Sought aesthetic dental treatment | | Received aesthetic dental treatment | |
| Characteristic | | | z test | | z test | | z test | | z test |
| | | % (95%CI) | p-value[#] | % (95%CI) | p-value[#] | % (95%CI) | p-value[#] | % (95%CI) | p-value[#] |
| **Sex** | | | | | | | | | |
| Male | | 34.7 (31.5–37.9) | <0.001* | 33.5 (30.3–36.7) | 0.001* | 70.6 (68.0–73.2) | <0.001* | 60.2 (57.4–63.0) | <0.001* |
| Female | | 42.5 (40.6–44.4) | | 40.0 (38.1–41.9) | | 81.1 (79.5–82.7) | | 70.8 (70.3–71.3) | |
| **Monthly income** | | | | | | | | | |
| Finland (€) | Brazil (R$) | | | | | | | | |
| <2,500 | <1,255 | 41.6 (39.2–44.0) | 0.218 | 39.4 (37.0–41.8) | 0.246 | 73.2 (67.7–78.7)[a] | <0.001* | 51.2 (45.0–57.4)[a] | <0.001* |
| 2,500 ⊢ 5,000 | 1,255 ⊢ 2,005 | 39.0 (35.7–42.3) | | 37.3 (34.0–40.6) | | 76.8 (72.7–80.9)[ab] | | 60.9 (56.2–65.6)[b] | |
| 5,000 ⊢ 7,500 | 2,005 ⊢ 8,641 | 39.6 (35.4–43.8) | | 37.7 (33.5–41.9) | | 79.6 (77.7–81.5)[b] | | 68.3 (66.2–70.4)[c] | |
| 7,500 ⊢ 10,000 | 8,641 ⊢ 11,262 | 38.8 (33.5–44.1) | | 35.9 (30.6–41.2) | | 81.5 (78.5–84.5)[b] | | 75.7 (72.4–79.0)[d] | |
| ≥10,000 | ≥11,262 | 41.9 (36.7–47.1) | | 39.6 (34.4–44.8) | | 73.6 (70.6–76.6)[a] | | 67.7 (64.5–70.9)[c] | |
| **Age (years)** | | | | | | | | | |
| <23 | | 42.0 (38.4–45.6) | 0.241 | 39.6 (36.0–43.2) | 0.324 | 80.6 (77.9–83.3)[c] | <0.001* | 68.4 (65.3–71.5)[b] | 0.006* |
| 23 ⊢ 29 | | 41.4 (38.7–44.1) | | 38.7 (36.0–41.4) | | 79.9 (77.3–82.5)[bc] | | 67.9 (64.9–70.9)[b] | |
| 29 ⊢ 39 | | 39.0 (35.5–42.5) | | 37.1 (33.6–40.6) | | 79.0 (76.6–81.4)[bc] | | 69.2 (66.5–71.9)[b] | |
| 39 ⊢ 52 | | 39.8 (35.7–43.9) | | 37.7 (33.6–41.8) | | 75.8 (72.5–79.1)[b] | | 66.7 (63.1–70.3)[ab] | |
| ≥52 | | 39.2 (34.0–44.4) | | 38.0 (32.8–43.2) | | 68.7 (64.3–73.1)[a] | | 61.9 (57.3–66.5)[a] | |

[#]the value presented is the lowest p-value found in pairwise comparison using z test ($\alpha = 5\%$) between the categories of the variable of interest.

*$p < 0.05$.

[ab]different letters indicate significant statistical difference.

satisfaction. The model showed an explained variance for life satisfaction of 9.8% in the Finnish sample and 13.1% for the Brazilian sample.

## Discussion

This study presents a screening of the prevalence of individuals seeking and undergoing esthetic dental treatment according to sociodemographic characteristics in Finnish and Brazilian population. Although an increase in demand for this treatment has been pointed out [7–10], there is a lack of specific data that allows comparisons across populations and sociodemographic groups. We also compared the self-perception of orofacial appearance according to sociodemographic characteristics and estimated its impact on subjective well-being in both populations. The results call attention to the importance of dental and medical practitioners, educators, and policy makers to know and deal with the sociodemographic and cultural aspects involved in aesthetic dental treatments.

Women were more likely than men to seek and undergo aesthetic dental treatment in both samples. This is expected, since Western cultures are marked by patriarchy roots [41] and objectification of female body [42], resulting in more body-altering behaviors by women [42]. However, the scenarios in Brazil and Finland are different. Finland has strong and effective gender equality policies in different spheres of life, such as access to education and health, paid work, and political empowerment [43, 44] and it is the second most gender-equal country in the world [22]. Nevertheless, our results corroborate previous studies [44, 45] showing that

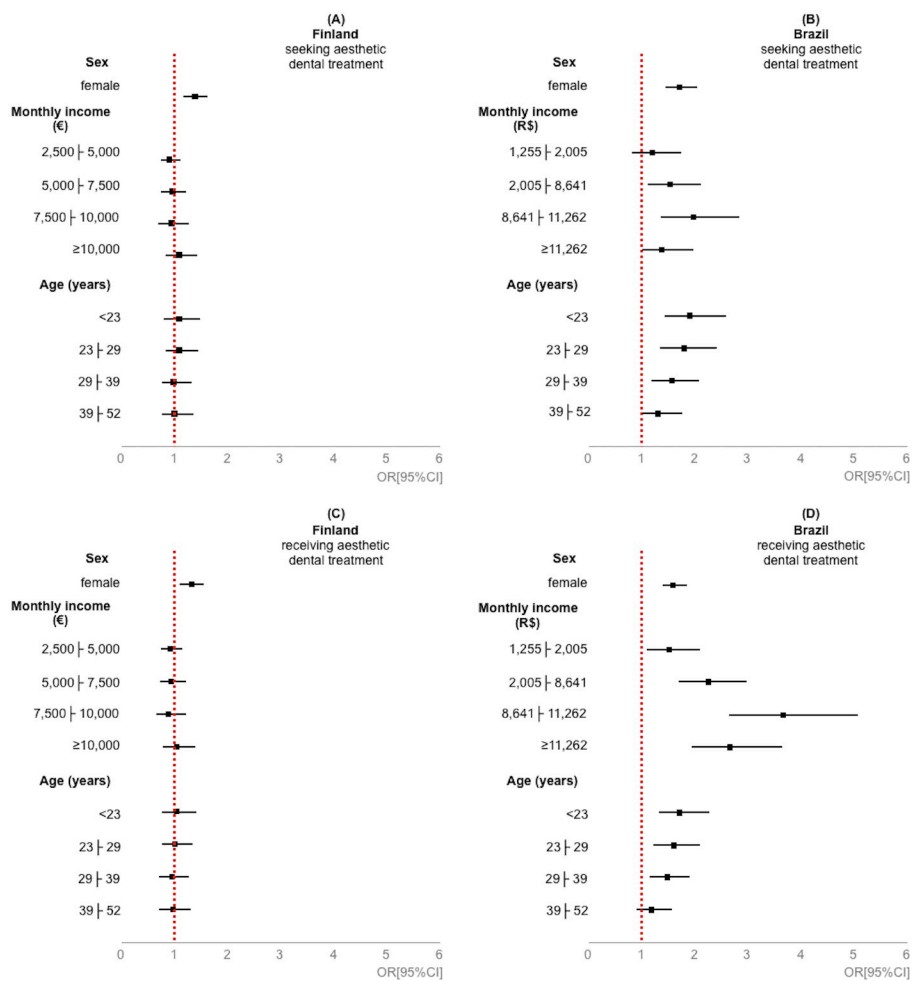

**Fig 2. Odds ratio (OR) with 95% confidence Interval (95%CI) for seeking and receiving aesthetic dental treatment according to sex, monthly income, and age. Note.** Reference category: sex = male; monthly income = <2,500€ / <R $1,255; and age = ≥52 years. Logistic Regression Models ($y_1$ = seeking aesthetic dental treatment; $y_2$ = receiving seeking aesthetic dental treatment; $X_1$ = sex; Monthly income (MI): $X_{MI}$, 2 = 2,500 ⊦5,000€/R$1,255 ⊦2,005; 3 = 5,000 ⊦7,500€/R$2,005 ⊦8,641; 4 = 7,500 ⊦10,000€/R$8,641 ⊦11,262; 5 = ≥10,000€/≥R$11,262; Age (years): $X_A$, 1 = <23; 2 = 23 ⊦29; 3 = 29 ⊦39; 4 = < 39 ⊦52): A: $y_1$ = -0.668+0.332$X_1$-0.081$X_{MI2}$-0.037$_{MI3}$-0.051$X_{MI4}$+0.087$X_{MI5}$+0.092$X_{A1}$+0.093$X_{A2}$-0.004$X_{A3}$+0.011$X_{A4}$, B: $y_1$ = 0.049 +0.551$X_1$+0.198$X_{MI2}$+0.444$_{MI3}$+0.688$X_{MI4}$+0.339$X_{MI5}$+0.655$X_{A1}$+0.596$X_{A2}$+0.463$X_{A3}$+0.288$X_{A4}$, C: $y_2$ = -0.669 +0.285$X_1$-0.072$X_{MI2}$-0.045$_{MI3}$-0.106$X_{MI4}$+0.054$X_{MI5}$+0.049$X_{A1}$+0.017$X_{A2}$-0.038$X_{A3}$-0.020$X_{A4}$, D: $y_2$ = -0.780 +0.472$X_1$+0.424$X_{MI2}$+0.816$_{MI3}$+1.303$X_{MI4}$+0.986$X_{MI5}$+0.547$X_{A1}$+0.480$X_{A2}$+0.399$X_{A3}$+0.180$X_{A4}$.

social norms evoking greater aesthetic pressure on women persist even in Finland. In future discussions and reform on gender equality policies in Finland [43] it is necessary to include agendas aimed at reconstructing the still-imposed social norms of physical appearance between the sexes.

In Brazil, with few and not so effective gender equality policies, there are social inequalities between the sexes [22], including low social representation of women in the society, wage gaps between the genders, and a high rate of violence against women [22, 46]. In this context, the greater physical appearance pressure on women, besides being considered a social norm, also becomes a tool for the men dominance and maintenance of inequalities [44]. Therefore, mini-mizing this pressure can be relevant, however, first and foremost, it is important that strong

**Table 3. Comparison of the mean scores (standard deviation) for each dimension of the Orofacial Esthetic Scale (OES) and Psychosocial Impact of Dental Aesthetic Questionnaire (PIDAQ) according to sex, monthly income, and age in the Finnish and Brazilian samples.**

| Country | Characteristic | Dimension[#] | | | | |
|---|---|---|---|---|---|---|
| | | OES | PIDAQ | | | |
| | | SOA | DSC | SI[‡] | PI[¶] | AC |
| Finland | **Sex** | | | | | |
| | Male | 6.9 (1.6) | 1.9 (1.0) | 0.5 (0.8) | 0.6 (0.7) | 0.6 (0.9) |
| | Female | 7.0 (1.6) | 2.0 (1.0) | 0.6 (0.8) | 0.9 (0.9) | 0.7 (1.0) |
| | Statistic test[†] | F = 2.31 | $F_W$ = 2.36 | $F_W$ = 11.34 | $F_W$ = 94.08 | $F_W$ = 3.97 |
| | p-value | 0.128 | 0.142 | 0.001* | <0.001* | 0.047* |
| | $\eta_p^2$ | 0.001 | 0.001 | 0.003 | 0.022 | 0.001 |
| | **Monthly income (€)** | | | | | |
| | <2,500 | 6.8 (1.6)[a] | 1.9 (1.0) | 0.7 (0.9)[b] | 1.0 (0.9)[b] | 0.8 (1.0)[b] |
| | 2,500 ⊢ 5,000 | 7.1 (1.6)[b] | 2.0 (1.0) | 0.5 (0.7)[a] | 0.8 (0.8)[a] | 0.7 (0.9)[a] |
| | 5,000 ⊢ 7,500 | 7.1 (1.6)[b] | 2.0 (1.1) | 0.5 (0.7)[a] | 0.8 (0.8)[a] | 0.7 (0.9)[ab] |
| | 7,500 ⊢ 10,000 | 7.2 (1.4)[b] | 2.0 (1.0) | 0.4 (0.6)[a] | 0.7 (0.7)[a] | 0.6 (0.8)[a] |
| | ≥10,000 | 7.4 (1.3)[b] | 2.1 (1.0) | 0.4 (0.7)[a] | 0.7 (0.8)[a] | 0.5 (0.9)[a] |
| | Statistic test[†] | $F_W$ = 15.62 | F = 2.15 | $F_W$ = 18.47 | $F_W$ = 16.06 | $F_W$ = 8.93 |
| | p-value | <0.001* | 0.072 | 0.001* | <0.001* | <0.001* |
| | $\eta_p^2$ | 0.016 | 0.002 | 0.020 | 0.017 | 0.009 |
| | **Age (years)** | | | | | |
| | <23 | 6.9 (1.6)[a] | 2.0 (1.0) | 0.7 (0.9)[b] | 1.0 (0.9)[b] | 0.7 (1.0)[b] |
| | 23 ⊢ 29 | 6.9 (1.6)[a] | 1.9 (1.0) | 0.6 (0.9)[b] | 1.0 (0.9)[b] | 0.8 (1.0)[b] |
| | 29 ⊢ 39 | 7.0 (1.6)[ab] | 1.9 (1.0) | 0.6 (0.8)[b] | 0.9 (0.9)[b] | 0.7 (0.9)[ab] |
| | 39 ⊢ 52 | 7.3 (1.5)[c] | 2.0 (1.1) | 0.4 (0.7)[a] | 0.7 (0.7)[a] | 0.6 (0.8)[a] |
| | ≥52 | 7.2 (1.6)[bc] | 1.9 (1.1) | 0.4 (0.6)[a] | 0.7 (0.7)[a] | 0.6 (0.8)[a] |
| | Statistic test[†] | F = 7.03 | $F_W$ = 1.31 | $F_W$ = 18.07 | $F_W$ = 18.45 | $F_W$ = 7.22 |
| | p-value | <0.001* | 0.266 | 0.001* | <0.001* | <0.001* |
| | $\eta_p^2$ | 0.008 | 0.001 | 0.016 | 0.017 | 0.007 |
| Brazil | **Sex** | | | | | |
| | Male | 7.1 (1.7) | 2.0 (1.0) | 0.5 (0.7) | 0.9 (0.9) | 0.8 (1.0) |
| | Female | 7.1 (1.7) | 2.0 (1.1) | 0.6 (0.8) | 1.1 (1.0) | 0.8 (1.0) |
| | Statistic test | F = 0.05 | F = 3.30 | $F_W$ = 8.44 | $F_W$ = 11.46 | $F_W$ = 2.43 |
| | p-value[†] | 0.816 | 0.069 | 0.006* | 0.001* | 0.119 |
| | $\eta_p^2$ | <0.001 | 0.001 | 0.002 | 0.003 | 0.001 |
| | **Monthly income (R$)** | | | | | |
| | <1,255 | 6.2 (2.1)[a] | 1.5 (1.1)[a] | 1.2 (1.1)[e] | 1.7 (1.2)[e] | 1.3 (1.2)[d] |
| | 1,255 ⊢ 2,005 | 6.7 (2.0)[b] | 1.8 (1.1)[b] | 0.9 (1.0)[d] | 1.4 (1.2)[d] | 1.1 (1.2)[c] |
| | 2,005 ⊢ 8,641 | 7.1 (1.7)[c] | 2.0 (1.1)[c] | 0.6 (0.8)[c] | 1.1 (1.0)[c] | 0.8 (1.0)[b] |
| | 8,641 ⊢ 11,262 | 7.3 (1.6)[d] | 2.1 (1.0)[d] | 0.4 (0.7)[b] | 0.8 (0.9)[b] | 0.7 (0.9)[ab] |
| | ≥11,262 | 7.5 (1.5)[d] | 2.2 (1.0)[d] | 0.3 (0.5)[a] | 0.7 (0.8)[a] | 0.6 (0.8)[a] |
| | Statistic test[†] | $F_W$ = 34.36 | F = 28.47 | $F_W$ = 71.28 | $F_W$ = 67.60 | $F_W$ = 31.41 |
| | p-value | 0.001* | <0.001* | <0.001* | <0.001* | <0.001* |
| | $\eta_p^2$ | 0.039 | 0.028 | 0.077 | 0.070 | 0.035 |
| | **Age (years)** | | | | | |
| | <23 | 6.8 (1.7)[a] | 1.8 (1.1)[a] | 0.9 (0.9)[d] | 1.4 (1.1)[c] | 1.0 (1.1)[b] |
| | 23 ⊢ 29 | 7.1 (1.7)[b] | 2.0 (1.1)[b,c] | 0.6 (0.8)[c] | 1.1 (1.0)[b] | 0.8 (1.0)[a] |
| | 29 ⊢ 39 | 7.3 (1.7)[b] | 2.1 (1.0)[c] | 0.5 (0.7)[b] | 0.9 (0.9)[a] | 0.7 (1.0)[a] |
| | 39 ⊢ 52 | 7.3 (1.8)[b] | 2.1 (1.1)[c] | 0.4 (0.7)[a] | 0.8 (0.9)[a] | 0.7 (0.9)[a] |

*(Continued)*

**Table 3.** (*Continued*)

| Country | Characteristic | Dimension[#] | | | | |
|---|---|---|---|---|---|---|
| | | OES | PIDAQ | | | |
| | | SOA | DSC | SI[‡] | PI[¶] | AC |
| | ≥52 | 7.1 (1.9)[b] | 1.9 (1.1)[a,b] | 0.4 (0.7)[a,b] | 0.9 (0.9)[a] | 0.8 (1.0)[a] |
| | Statistic test[†] | $F_W = 13.98$ | F = 10.92 | $F_W = 37.37$ | $F_W = 39.20$ | $F_W = 10.07$ |
| | p-value | <0.001* | <0.001* | <0.001* | <0.001* | <0.001* |
| | $\eta_p^2$ | 0.014 | 0.011 | 0.018 | 0.042 | 0.011 |

[#]SOA: Satisfaction with Orofacial Appearance; DSC: Dental Self-Confidence; SI: Social Impact; PI: Psychological Impact; AC: Aesthetic Concern.

[†]F: ANOVA; $F_w$: Welch's ANOVA.

*$p < 0.05$.

[ab]different letters indicate significant statistical difference among groups according to dimension (Tukey or Games-Howell post hoc test, $\alpha$ = 5%).

[‡]For Finnish sample, items 9, 13, 14 and 15 were not considered for the calculation of the score.

[¶]For Brazilian sample, item 6 was not considered for the calculation of the score.

and effective policies are developed and promoted to build a safe, fair, and representative society for Brazilian women as well.

No or very small differences without practical significance in the self-perception of the orofacial appearance (assessed by OES and PIDAQ) were observed between men and women.

**Table 4. Path estimates of the structural models elaborated to assess the impact of orofacial appearance on life satisfaction, the moderation role of sex, monthly income, and age, and the indirect effect of have received aesthetic dental treatment on this impact.**

| Path estimate | Finnish sample | | | | Brazilian sample | | | |
|---|---|---|---|---|---|---|---|---|
| | B | β | SE | p-value | B | β | SE | p-value |
| **Completed model** | | | | | | | | |
| OA → LS (β1) | 0.52 | 0.45 | 0.16 | 0.002 | 0.47 | 0.44 | 0.10 | <0.001 |
| ADT → LS (β2) | 0.04 | 0.01 | 0.04 | 0.279 | 0.13 | 0.05 | 0.04 | 0.002 |
| Sex → LS (β3) | 0.33 | 0.11 | 0.05 | <0.001 | 0.12 | 0.04 | 0.02 | 0.007 |
| MI → LS (β4) | 0.21 | 0.22 | 0.02 | <0.001 | 0.37 | 0.30 | 0.02 | <0.001 |
| Age → LS (β5) | -0.01 | -0.07 | <0.01 | <0.001 | 0.01 | 0.08 | <0.01 | <0.001 |
| Sex*OA → LS (β6) | -0.12 | -0.18 | 0.07 | 0.099 | -0.07 | -0.11 | 0.04 | 0.101 |
| MI*OA → LS (β7) | -0.01 | -0.03 | 0.02 | 0.523 | -0.02 | -0.06 | 0.02 | 0.159 |
| Age*OA → LS (β8) | <0.01 | 0.03 | <0.01 | 0.621 | <0.01 | -0.01 | <0.02 | 0.790 |
| OA → ADT (β9) | -0.01 | -0.04 | 0.01 | 0.030 | 0.01 | 0.03 | 0.01 | 0.104 |
| Sex → ADT (β10) | 0.06 | 0.05 | 0.02 | 0.001 | 0.11 | 0.10 | 0.02 | <0.001 |
| MI → ADT (β11) | <0.01 | <0.01 | 0.01 | 0.845 | 0.05 | 0.12 | 0.01 | <0.001 |
| Age → ADT (β12) | <0.01 | <0.01 | <0.01 | 0.995 | <0.01 | -0.08 | <0.01 | <0.001 |
| *Indirect effect*[#] | | | | | | | | |
| OA → ADT → LS (β9*β2) | <0.01 | <0.01 | <0.01 | 0.384 | <0.01 | <0.01 | <0.01 | 0.166 |
| **Refined model**[†] | | | | | | | | |
| OA → LS (β1) [‡] | 0.34 | 0.31 | 0.02 | <0.001 | 0.38 | 0.36 | 0.02 | <0.001 |

B: non-standardized path estimate; β: standardized path estimate; SE: standard error; OA: orofacial appearance dimension; ADT: have received aesthetic dental treatment; LS: life satisfaction; MI: monthly income. β1 to β12: path estimates corresponding to Fig 1B.

[#]Indirect effect assessed by Sobel's test with bootstrap simulation.

[†]Model refined by excluding the variables have received aesthetic dental treatment, sex, monthly income, and age (Fig 1C).

[‡]Explained variance for life satisfaction: Finnish sample = 0.098, Brazilian sample = 0.131.

These results refute that the higher demand by women occurs because of a poorer self-perception of orofacial appearance than men. Rather, they support the idea that external factors and the internalization of social norms [41, 42, 44, 45], have a significant contribution in the difference in seeking aesthetic dental treatment between the sexes.

In agreement with previous studies [16, 20, 21], no differences with practical significance were found in the perception of orofacial appearance between the age groups. Despite this, in Brazil, young people have sought and undergone more aesthetic dental treatments. An explanation for this is that Brazilians place a high value on appearance and it becomes a key component of social interactions [1, 2] and can be considered as a capital (aesthetic capital) [47]. In this regard, good looks based on beauty standards have a strong influence on one's social acceptance and on obtaining job positions [2]. Therefore, young Brazilians may have a high demand for aesthetic dental treatment to fit into socially established beauty standards with the aim of achieving social insertion and even professional position.

In Finland, although physical appearance also has an influence on many aspects of life [44, 47, 48], no difference in the demand for aesthetic dental treatment was observed between the age groups. This difference in relation to Brazil can be interpreted by speculating three different, but not mutually exclusive, hypotheses: 1. a lower value attributed to physical appearance in Finland than in Brazil. It may result in lower demand for aesthetic dental treatment, even though the perception of orofacial appearance is similar to the Brazilians [12]. 2. different beauty standards related to physical traits between countries. In Brazil, the ideal beautiful smile is close to that of the USA [49], which includes straight and very white teeth and explains the greater demand for smile-improving treatments. And 3. in Finland there is a social norm of equal opportunity for the population [48], so that Finns may be aware of discriminatory issues that violate it, such as those related to physical appearance [48]. In this way, many Finns may avoid adopting behaviors, including undergoing aesthetic treatment, that contribute to these issues.

Despite the differences between Brazil and Finland, it is important to point out that aesthetic values, beauty standards, and behaviors to alter physical appearance have changed and become more similar among countries, especially Western ones, with increasing digitalization and the rise of social media [50]. Therefore, future cross-national studies in different age cohorts, including younger generations, are important for understanding attitudes toward orofacial appearance and aesthetic treatment. Our results may also not have captured the age-related consumption of aesthetic treatments for the rejuvenation purpose. This is because only intraoral treatments were considered as aesthetic dental treatment in the present study. Although some intraoral clinical features are associated with a more youthful appearance (e.g., gingival display and shape and length of incisors) [51], most treatments aiming at rejuvenation effect are extraoral, such as botulinum toxin and soft tissue filler injections [52]. Thus, we suggest that future studies investigate the relation of demand for different aesthetic treatments with self-perception of appearance in different populations and groups.

The results regarding monthly income were also different between Finland and Brazil. It supports the idea that health treatments, especially aesthetic ones, can be associated with consumerism and social prestige [12, 53]. Finland is classified as a low socioeconomic inequality country between socioeconomic classes [23] and most Finns have similar living conditions and a democratic access to health care, regardless of socioeconomic classes. Brazil, in turn, is classified as a high-inequality country with regard to socioeconomic factors [23], including differences in the access to healthcare. Still, treatments solely or primarily intended to improve physical appearance, such as aesthetic dental treatments, are not offer by the Brazilian public health system and can only be accessed in private clinics at high cost. Our results show that Brazilians in middle and upper socioeconomic classes had more access to aesthetic dental

treatment than the lower class, emphasizing the importance of income as a key factor in seeking and accessing dental treatments. Therefore, it is essential to consider economic conditions when studying the conditions that intervene in the search and undergoing of dental treatments.

The results also reinforce the hypothesis that these treatments have high social and consumption values in countries with large social inequalities, such as Brazil [12]. It is also difficult to disassociate these values from the other results, which show that lower-income Brazilians had a higher negative social and psychological impact of dental aesthetics on their life. The relation of this impact with not having undergone aesthetic dental treatment may be associated with a dissatisfaction with a physical characteristic, as well as with an unfulfilled desire to consume and a consequent feeling of not belonging to higher socioeconomic classes.

In accordance with previous studies [1, 54, 55], in which physical appearance was found to be an important contribution to one's subjective well-being, our results showed that self-perception of orofacial appearance had a significant impact on life satisfaction. This perception contributed approximately one tenth to the life satisfaction of Brazilian and Finnish individuals. Having received aesthetic dental treatment did not have an indirect effect on this impact.

Nevertheless, these results demonstrate how powerful the performance of the dentist can be in the re-establishment and/or promotion of their patient's well-being. In some cases, the aesthetic dental treatments are well indicated, with an improvement of physical aspects and, consequently, of the self-perception of orofacial appearance. This may, in turn, provide psychological benefits and positively impacts the patient's life satisfaction. However, this effect may not have been captured in our study since the sample consists of individuals from the general population, rather than patients with a specific orofacial condition. Thus, the results also suggest that the demand for aesthetic dental treatment is not always solely motivated by a desire to improve a single physical aspect, pointing to the potential risks that treatments carried out indiscriminately and without individualized planning may have. This is because, in some cases, the demand for aesthetic dental treatment may be associated with psychological symptoms or disorders (e.g., dysmorphic disorder) [56] or social pressures as discussed above. For these, performing the aesthetic treatment may not have a long-term benefit [1, 56] and may also contribute to worsening psychological symptoms or disorders and to the maintenance of social pressures and inequalities. As a result, no benefit, or even a negative impact on the patient's well-being, may be observed.

Present structural model analysis indicated that the impact of orofacial appearance on life satisfaction was similar between the countries and was not moderated by sociodemographic characteristics. This is comprehensible because orofacial region has peculiarities that transcend time, culture, and sociocultural characteristics [4]. Orofacial region not only serves as a tool for communication (both verbal and nonverbal) [1, 3], but also plays a crucial role in shaping one's sense of self-identity through unique physical features [1, 4]. Keeping in mind the importance of individual needs and characteristics, aesthetic treatments in the orofacial region should be patient-centered including a detailed anamnesis, patient's perceptions, and clinical examination identifying unique characteristics. In this way, the treatment can address physical issues that may enhance the sense of belonging and connect the patient to their social/cultural group [12, 53]. At the same time, the treatment will preserve their singularities, maintaining the sense of individuality and uniqueness [4]. Otherwise, the individual's singular characteristics are not taken into account in the aesthetic treatment, often being altered or disguised. This alteration may negatively affect the patient by removing their sense of self-identity and lead to a lack of recognition of themselves [4], subsequently causing dissatisfaction with the treatment.

The cross-sectional design was a limitation of the study since it does not allow for cause-and-effect inference of the structural model. The non-probability sampling and the online data collection can also be considered as a limitation [12], as they may hinder the generalizability of the results to the whole Finnish and Brazilian population. Trying to minimize these limitations, we used large sample sizes to obtain a comprehensive result that is close to the variability of the study population. We also attested to the validity and reliability of the data and used robust methods to elaborate the structural models. It is also noteworthy that the present study used life satisfaction as a measure of well-being. It deals only with the cognitive aspect of well-being from a hedonic perspective (focused on experiences of pleasure and enjoyment) [27–29]. However, well-being is a multidimensional concept, and other examples of its aspects are emotional well-being (hedonic perspective) and social and psychological well-being (eudaimonic perspective: focused on experiences of meaning and purpose) [55]. Therefore, it is important for future studies to examine the relationship between self-perception of physical appearance and other aspects of well-being in different cultures.

Despite its limitations, the present study provides evidence that contributes to research on social determinants of health, include those conducted in Latin America [57, 58]. It also highlights the need for discussion and further investigation of the psychological, social, cultural, economic, and political factors related to aesthetic dental treatments and their implications for health across different countries. These efforts can identify and provide key elements for a better understanding of social determinants of health, which is crucial for the development of effective and equitable health policies and programs. This can lead to improved health outcomes for individuals and communities.

## Conclusion

The demand for aesthetic dental treatment is influenced by sociodemographic and cultural factors, not just by self-perception of orofacial appearance. The findings indicate greater societal pressure on physical appearance among women in Finland and Brazil. They also suggest that consumerism and social prestige are involved in this demand in countries with high socio-economic inequalities, such as Brazil. Self-perception of orofacial appearance plays a significant role in individuals' subjective well-being. Therefore, to achieve success and promote well-being, the planning of aesthetic treatments in the orofacial region should also take into account the patient's perspectives, perceptions, unique characteristics, and social context.

## Supporting information

**S1 Table. Psychometric indicators related to the fit of the factor models of Orofacial Esthetic Scale (OES), Psychosocial Impact of Dental Aesthetics Questionnaire (PIDAQ), and Satisfaction with Life Scale (SWLS) to the samples.**
(DOCX)

**S2 Table. Descriptive statistics of the scores of Orofacial Esthetic Scale (OES) and Psychosocial Impact of Dental Aesthetic Questionnaire (PIDAQ) dimensions and measures of sample adequacy (MSA) for principal component analysis (Finnish sample: n = 3,614; Brazilian sample: n = 3,979).**
(DOCX)

**S3 Table. Principal component analysis (PCA) and parallel analysis results for Finnish and Brazilian sample.**
(DOCX)

**S1 File. Data.** Data underlying the finds described in this manuscript.
(XLSX)

## Author Contributions

**Conceptualization:** Lucas Arrais Campos, Juliana Alvares Duarte Bonini Campos, Timo Peltomäki.

**Data curation:** Lucas Arrais Campos, Juliana Alvares Duarte Bonini Campos, Timo Peltomäki.

**Formal analysis:** Lucas Arrais Campos, Juliana Alvares Duarte Bonini Campos, João Marôco.

**Funding acquisition:** Lucas Arrais Campos, Juliana Alvares Duarte Bonini Campos, João Marôco, Timo Peltomäki.

**Investigation:** Lucas Arrais Campos, Juliana Alvares Duarte Bonini Campos, Timo Peltomäki.

**Methodology:** Lucas Arrais Campos, Juliana Alvares Duarte Bonini Campos, João Marôco, Timo Peltomäki.

**Project administration:** Juliana Alvares Duarte Bonini Campos, Timo Peltomäki.

**Resources:** Lucas Arrais Campos, Juliana Alvares Duarte Bonini Campos, Timo Peltomäki.

**Software:** Juliana Alvares Duarte Bonini Campos, João Marôco.

**Supervision:** Juliana Alvares Duarte Bonini Campos, João Marôco, Timo Peltomäki.

**Validation:** Lucas Arrais Campos, Juliana Alvares Duarte Bonini Campos.

**Visualization:** Lucas Arrais Campos, Timo Peltomäki.

**Writing – original draft:** Lucas Arrais Campos.

**Writing – review & editing:** Juliana Alvares Duarte Bonini Campos, João Marôco, Timo Peltomäki.

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
