## [Decision Letter · Decision Letter 0]

25 Apr 2023

PONE-D-23-06321Aesthetic dental treatment, orofacial appearance, and life satisfaction of Finnish and Brazilian adultsPLOS ONE

Dear Dr. Campos,

Thank you for submitting your manuscript to PLOS ONE. After careful consideration, we feel that it has merit but does not fully meet PLOS ONE’s publication criteria as it currently stands. Therefore, we invite you to submit a revised version of the manuscript that addresses the points raised during the review process.

We look forward to receiving your revised manuscript.

Kind regards,

Ana Cristina Mafla

Academic Editor

PLOS ONE

Journal Requirements:

“LAC (FAPESP #2018/06739-1 and CAPES Finance Code 001): PhD scholarship

JADBC (FAPESP #2019/19590-9): research grant

FAPESP: São Paulo Research Foundation; https://fapesp.br/en

CAPES: Coordenação de Aperfeiçoamento de Pessoal de Nível Superior - Brasil; https://www.gov.br/capes/pt-br”

Please include this amended Role of Funder statement in your cover letter; we will change the online submission form on your behalf

Reviewers' comments:

Reviewer's Responses to Questions

**Comments to the Author**

1. Is the manuscript technically sound, and do the data support the conclusions?

Reviewer #1: Yes

Reviewer #2: Yes

2. Has the statistical analysis been performed appropriately and rigorously? 

Reviewer #1: Yes

Reviewer #2: Yes

3. Have the authors made all data underlying the findings in their manuscript fully available?

Reviewer #1: Yes

Reviewer #2: Yes

4. Is the manuscript presented in an intelligible fashion and written in standard English?

Reviewer #1: Yes

Reviewer #2: Yes

5. Review Comments to the Author

Reviewer #1: 

The subject of this manuscript is of interest for research in several fields such as body image, mental health, and dentistry, as also for clinical practice. This is a well-written manuscript resulting from a robust method. Reading the text was very informative. I really enjoyed it. Below I present a few comments regarding some information I felt was lacking and some aspects that raised my curiosity.

Introduction:

1. Some background on particular characteristics of Brazil and Finnish that might impact the studied variables might be informative for the readers. These countries are very different regarding the value placed on physical appearance, the access to health care, including dental treatments and aesthetic procedures, and the cost of the procedures – for example. Although some of these aspects were presented in the discussion section, I think it is important to state in the introduction what are the reasons (if any) to investigate these countries in a single study (and to compare them).

Method:

2. What were the criteria used to categorize the variable income?

3. Did the participants receive some incentive to participate in the study? Did they receive information about the results of the study?

4. Since it was an online survey using snowball sampling, did the authors employ some strategy to confirm that the respondents were adults?

5. When examing the validity of the data using confirmatory factor analysis, did the authors test data normality (both univariate and multivariate)?

6. For the analysis of logistic regression, how the reference categories of each variable were established?

Reviewer #2: 

This study is very interesting because it shows the importance of sociodemographic and psychological variables as predictors of the seek/undergoing aesthetic dental treatment. The sample size and the rigor of the statistical procedures represent a great strength of the work presented.

The document shows four specific objectives however, there is no general objective that allows the articulation of all of them in such a way that the final purpose of the study presented can be identified.

The fourth specific objective, "to conduct a cross-national study", is not essentially an objective, but rather an activity. It is already integrated into the specific objectives when the comparison between Brazilians and Finnish is mentioned.

When reviewing the coherence and integration between the objectives of the study, it is identified that the first one relates sociodemographic variables with the probability of seeking and undergoing aesthetic dental treatment; the second objective relates sociodemographic variables to facial appearance; and the third objective relates OFA to SWL. However, it would be important to explore if the sociodemographic variables, OFA, and SWL are related to the probability of undergoing aesthetic dental treatment. This can be an alternative to the articulation of specific objectives.

There is no clarity in the text of lines 70 and 71: "Individuals who were more satisfied and had less psychosocial impact from OA had higher levels of LS". What does the word "satisfied" refer to?

It is necessary to clarify the criteria that were used to create the income ranges. It would be necessary to check if the income issue is comparable, in terms of what is required to live. The fact that the majority of Finnish are in quartile 1 may indicate a guarantee of living conditions, while in Brazil it may show inequities in income distribution.

It is necessary to clarify in the text how the calculation of the effect size was made.

Access to treatment may be mediated by economic conditions. The difference in OR behavior between those who seek and those who undergo aesthetic dental treatment shows the importance of income as a key factor, reason for which it must be considered in the study of the conditions that intervene in the search and realization of dental treatments.

It is suggested to update Figure 3, which corresponds to the SEM, so that the final model evaluated is included.

Finally, it is important to emphasize that the study of the conditions that promote the development of specific health results has been a topic developed from Latin American models known as critical epidemiology or the social determination of health, proposals that can provide new elements of understanding of how the social, cultural, economic, and political conditions interact so that results such as those discussed in the study are presented.

In this regard, I respectfully suggest some references:

Breilh J. La determinación social de la salud como herramienta de transformación hacia una nueva salud pública (salud colectiva). Rev. Fac. Nac. Salud Pública 2013; 31(supl 1): S13-S27. Available: http://www.scielo.org.co/pdf/rfnsp/v31s1/v31s1a02.pdf

Ruiz DC, Morales C. Social determination of the oral health disease process: a social-historical approach in four Latin American countries. Invest Educ Enferm. 2015; 33(2): 248-259. Available: http://www.scielo.org.co/pdf/iee/v33n2/v33n2a07.pdf

Concha S. Determinación Social de la atención odontológica de las mujeres embarazadas de tres localidades de Bogotá. Tesis Doctoral. Bogotá: Universidad Nacional de Colombia. Available: https://repositorio.unal.edu.co/bitstream/handle/unal/55763/%281%2963317599.2015.pdf?sequence=1&isAllowed=y

6. PLOS authors have the option to publish the peer review history of their article (what does this mean?). If published, this will include your full peer review and any attached files.

Reviewer #1: No

Reviewer #2: No

---

## [Author Response · Author response to Decision Letter 0]

14 May 2023

Dear Dr. Ana Cristina Mafla,

Academic Editor 

PLOS ONE

Thank you for reviewing our manuscript and considering the study for publication after the requested review. We are submitting a revised manuscript highlighting the changes and a clean version. Please kindly see below our response point-by-point to reviewers’ comments and suggestions.

We hope that the responses and the revised manuscript address the reviewers' comments.

Sincerely,

The Authors

Reviewer #1

•The subject of this manuscript is of interest for research in several fields such as body image, mental health, and dentistry, as also for clinical practice. This is a well-written manuscript resulting from a robust method. Reading the text was very informative. I really enjoyed it. Below I present a few comments regarding some information I felt was lacking and some aspects that raised my curiosity. 

-Response: Thank you.

•Introduction:

1. Some background on particular characteristics of Brazil and Finnish that might impact the studied variables might be informative for the readers. These countries are very different regarding the value placed on physical appearance, the access to health care, including dental treatments and aesthetic procedures, and the cost of the procedures – for example. Although some of these aspects were presented in the discussion section, I think it is important to state in the introduction what are the reasons (if any) to investigate these countries in a single study (and to compare them). 

-Response: Thank you for your comment. We have added to the introduction some differences that justify the comparison between the countries (lines: 145-159) and also a sentence in the methods (lines: 172-174).

•Method:

2. What were the criteria used to categorize the variable income? 

-Response: The categories for the monthly income variable were defined based on the recommendations and criteria of Statistics Finland and Centro de Políticas Sociais - FGV Social (Brazil). Both institutions are involved in research aimed at informing societal debates and the implementation of public policies.

This information has been added to the manuscript (lines: 190-192).

•3. Did the participants receive some incentive to participate in the study? Did they receive information about the results of the study? 

-Response: No, participation in the study was voluntary and anonymous in both countries, and participants did not receive any incentives to take part (information added in lines: 214-215).

As data collection was anonymous and we did not have access to information that could identify individual participants during or after data collection, participants did not receive the study results directly. However, participants were provided with the researchers' contact information before and after the study to request additional information and the study results. Furthermore, we aim to disseminate our academic findings to the broader society using non-academic language. This is achieved through publication on the universities' institutional websites and through mainstream media, such as newspaper and magazine interviews. In Finland, results related to the findings have been discussed in two interviews for national journals and magazines. In Brazil, the results can be presented institutionally after the manuscript is published.

•4. Since it was an online survey using snowball sampling, did the authors employ some strategy to confirm that the respondents were adults? 

-Response: Thank you for your question. We adhered to the guidelines and regulations set by the Data Protection Officer in Finland and the Research Ethics Committee in Brazil for online data collection. The only measure we took to ensure that respondents were adults was to include a question about their age and obtain their consent to participate in the study, which was obtained only after informing participants about the nature of the research and its inclusion criteria for adult individuals.

•5. When examing the validity of the data using confirmatory factor analysis, did the authors test data normality (both univariate and multivariate)? 

-Response: Yes, the assumption of normality (both univariate and multivariate) was attested for the CFA. We have added this information and results to the manuscript (lines: 243-249 and 266-270).

•6. For the analysis of logistic regression, how the reference categories of each variable were established? 

-Response: The reference categories for the independent variables were established based on previous studies that identified the groups with the lowest prevalence of seeking/undergoing esthetic treatments (information added in lines: 287-289). 

Reviewer #2 

•This study is very interesting because it shows the importance of sociodemographic and psychological variables as predictors of the seek/undergoing aesthetic dental treatment. The sample size and the rigor of the statistical procedures represent a great strength of the work presented. 

-Response: Thank you.

•The document shows four specific objectives however, there is no general objective that allows the articulation of all of them in such a way that the final purpose of the study presented can be identified. 

-Response: Thank you for your comment. The addition of an objective integrating the relationship between orofacial appearance, satisfaction with life, demographic variables, and receiving aesthetic dental treatment undoubtedly brings coherence and improves the manuscript. Please see below for our modifications and additions.

•The fourth specific objective, "to conduct a cross-national study", is not essentially an objective, but rather an activity. It is already integrated into the specific objectives when the comparison between Brazilians and Finnish is mentioned. 

-Response: The reviewer is correct. Therefore, we have removed this fourth objective.

•When reviewing the coherence and integration between the objectives of the study, it is identified that the first one relates sociodemographic variables with the probability of seeking and undergoing aesthetic dental treatment; the second objective relates sociodemographic variables to facial appearance; and the third objective relates OFA to SWL. However, it would be important to explore if the sociodemographic variables, OFA, and SWL are related to the probability of undergoing aesthetic dental treatment. This can be an alternative to the articulation of specific objectives. 

-Response: Thank you for the comment. We appreciate it and agree that adding an objective that integrates sociodemographic variables, orofacial appearance, life satisfaction, and aesthetic treatment makes the manuscript more coherent.

To elaborate this new objective, we considered that the "dental aesthetic treatment" variable was obtained through a question asking whether the participant had received this treatment. We also noted that that "life satisfaction", measured by SWLS, refers to the overall life satisfaction at the time of filling out the survey (after the participant had undergone the aesthetic treatment). Thus, we have decided to keep life satisfaction as dependent variable, and include aesthetic dental treatment in the SEM.

Based on the data, we believe that the model that considers the indirect effect of the treatment variable on the impact of orofacial appearance on life satisfaction is the most coherent and theoretically plausible. This model also maintains moderation role of sociodemographic variables on this impact and includes a direct path from these variables to treatment. We remain open to any other suggestions the reviewer may have.

We have modified and/or added information in the objectives (lines: 56-57 and 163-166), methods (lines: 324-327, 335-337; Fig. 1), results (lines: 422-430; Table 4), and discussion (lines: 546-554).

•There is no clarity in the text of lines 70 and 71: "Individuals who were more satisfied and had less psychosocial impact from OA had higher levels of LS". What does the word "satisfied" refer to? 

-Response: We added information to make it clearer (“Individuals who were more satisfied with their own OA…”; line: 72)

•It is necessary to clarify the criteria that were used to create the income ranges. It would be necessary to check if the income issue is comparable, in terms of what is required to live. The fact that the majority of Finnish are in quartile 1 may indicate a guarantee of living conditions, while in Brazil it may show inequities in income distribution. 

-Response: The categories for the monthly income variable were defined based on the recommendations and criteria of Statistics Finland and Centro de Políticas Sociais - FGV Social (Brazil). Both institutions are involved in research aimed at informing societal debates and the implementation of public policies considering the reality of each country. This information has been added to the manuscript (lines: 190-192).

Following Reviewer 1's comment, we have added a paragraph to the introduction highlighting the differences between the countries, including differences in living conditions and inequalities among socioeconomic classes (lines: 150-155). Additionally, we have included in the text that the results point to these differences and that income categories are not directly comparable (lines: 355-360).

•It is necessary to clarify in the text how the calculation of the effect size was made. 

-Response: The calculation of effect sizes has been added to the Methods (lines: 300-301 and 341). Thank you.

•Access to treatment may be mediated by economic conditions. The difference in OR behavior between those who seek and those who undergo aesthetic dental treatment shows the importance of income as a key factor, reason for which it must be considered in the study of the conditions that intervene in the search and realization of dental treatments. 

-Response: Thank you for the comment. We add some sentences in discussion to reinforce this idea of the importance of income as a key factor in seeking and accessing dental treatments (lines: 529-533).

•It is suggested to update Figure 3, which corresponds to the SEM, so that the final model evaluated is included. 

-Response: Figure 1 has been updated according to the new proposed aim and adding the final refined model.

•Finally, it is important to emphasize that the study of the conditions that promote the development of specific health results has been a topic developed from Latin American models known as critical epidemiology or the social determination of health, proposals that can provide new elements of understanding of how the social, cultural, economic, and political conditions interact so that results such as those discussed in the study are presented.

In this regard, I respectfully suggest some references:

Breilh J. La determinación social de la salud como herramienta de transformación hacia una nueva salud pública (salud colectiva). Rev. Fac. Nac. Salud Pública 2013; 31(supl 1): S13-S27. Available: http://www.scielo.org.co/pdf/rfnsp/v31s1/v31s1a02.pdf

Ruiz DC, Morales C. Social determination of the oral health disease process: a social-historical approach in four Latin American countries. Invest Educ Enferm. 2015; 33(2): 248-259. Available: http://www.scielo.org.co/pdf/iee/v33n2/v33n2a07.pdf

Concha S. Determinación Social de la atención odontológica de las mujeres embarazadas de tres localidades de Bogotá. Tesis Doctoral. Bogotá: Universidad Nacional de Colombia. Available: https://repositorio.unal.edu.co/bitstream/handle/unal/55763/%281%2963317599.2015.pdf?sequence=1&isAllowed=y

-Response: Thank you for the comment. We have added a final paragraph to the discussion, as well as the suggested references that we had access, emphasizing the contribution of our study and the importance of investigating this topic. (lines: 595-602)

---

## [Editor Report · Decision Letter 1]

2 Jun 2023

Aesthetic dental treatment, orofacial appearance, and life satisfaction of Finnish and Brazilian adults

PONE-D-23-06321R1

Dear Dr. Lucas Arrais Campos,

We’re pleased to inform you that your manuscript has been judged scientifically suitable for publication and will be formally accepted for publication once it meets all outstanding technical requirements.

Kind regards,

Ana Cristina Mafla

Academic Editor

PLOS ONE

---

## [Editor Report · Acceptance letter]

21 Jun 2023

PONE-D-23-06321R1 

Aesthetic dental treatment, orofacial appearance, and life satisfaction of Finnish and Brazilian adults 

Dear Dr. Campos:

I'm pleased to inform you that your manuscript has been deemed suitable for publication in PLOS ONE. Congratulations! Your manuscript is now with our production department. 

Kind regards, 

on behalf of

Dr. Ana Cristina Mafla 

Academic Editor

PLOS ONE